# Metagenome mining and functional analysis reveal oxidized guanine DNA repair at the Lost City Hydrothermal Field

**Payton H. Utzman**, **Vincent P. Mays**, **Briggs C. Miller, Mary C. Fairbanks, William J. Brazelton, Martin P. Horvath***

School of Biological Sciences, University of Utah, Salt Lake City, Utah, United States of America

☯ These authors contributed equally to this work.
* martin.horvath@utah.edu

**Data Availability Statement:** Primary data and code for analysis of data have been deposited with GitHub: https://github.com/paytonutzman/Lost-City-MutY-Discovery. These include FASTA files for

## Abstract

The GO DNA repair system protects against GC → TA mutations by finding and removing oxidized guanine. The system is mechanistically well understood but its origins are unknown. We searched metagenomes and abundantly found the genes encoding GO DNA repair at the Lost City Hydrothermal Field (LCHF). We recombinantly expressed the final enzyme in the system to show MutY homologs function to suppress mutations. Microbes at the LCHF thrive without sunlight, fueled by the products of geochemical transformations of seafloor rocks, under conditions believed to resemble a young Earth. High levels of the reductant $H_2$ and low levels of $O_2$ in this environment raise the question, why are resident microbes equipped to repair damage caused by oxidative stress? MutY genes could be assigned to metagenome-assembled genomes (MAGs), and thereby associate GO DNA repair with metabolic pathways that generate reactive oxygen, nitrogen and sulfur species. Our results indicate that cell-based life was under evolutionary pressure to cope with oxidized guanine well before $O_2$ levels rose following the great oxidation event.

## Introduction

The Lost City Hydrothermal Field (LCHF) resembles conditions of a younger planet and thus provides a window to study the origin of life on Earth and other planets [1–3]. Located 15 kilometers from the Mid-Atlantic Ridge, the LCHF comprises a series of carbonate chimneys at depths ranging from 700–800 meters below the ocean surface [3]. The temperature and pH of the LCHF are both elevated, with temperature ranging from 40˚C to 90˚C and pH ranging from 9 to 11 [1]. At this depth, light does not penetrate, magmatic energy sources are unavailable, and dissolved carbon dioxide is scarce [1]. Despite these environmental constraints, archaea and bacteria inhabit the chimneys and hydrothermal fluids venting from the subsurface [1, 4]. The chemoautotrophic microbes take advantage of chemical compounds generated by subseafloor geochemical reactions such as serpentinization, the aqueous alteration of ultramafic rocks [1]. Serpentinization produces hydrogen gas and low-molecular-weight hydrocarbons, which fuel modern microbial communities and also would have been needed to fuel self-replicating molecules and the emergence of primitive metabolic pathways as an antecedent to

the discovered MutY enzymes, predicted structures, virtual docking outcomes, input files to generate the MD trajectories and scripts to analyze the MD trajectories, rifampicin resistance frequency data, and the R code to report statistics, median and 95% confidence intervals.

**Funding:** This work was supported by National Science Foundation (https://www.nsf.gov/) awards to Martin P. Horvath (CHE:CLP- 1905249, 2204229) and to William J. Brazelton. (OCE-1536405). This work was also supported by UROP funding from the Office of Undergraduate Research at the University of Utah (https://our.utah.edu/) to Payton H. Utzman. This work was also supported by the NASA Astrobiology Institute Rock-Powered Life team. The funders had no role in study design, data collection and analysis, decision to publish, or preparation of the manuscript. There was no additional external funding received for this study.

**Competing interests:** The authors have declared that no competing interests exist.

cellular life [1, 2, 5]. Hydrothermal circulation underneath the LCHF depletes seawater oxygen, leading to an anoxic hydrothermal environment very different from the nearby oxygen-rich seawater [1, 3, 6–8]. As such, the subsurface microbial communities may offer a glimpse into how life emerged and existed before the Great Oxidation Event that occurred over two billion years ago.

The unusual environmental conditions of the LCHF present several biochemical challenges to the survival of microbes [4, 7, 8]. For example, high temperatures and alkaline pH conditions present at the LCHF potentiate chemistry to generate DNA-damaging, reactive oxygen species [9]. However, it remains unclear whether reactive oxygen species (ROS) are a major threat to resident microbes given that the subseafloor underneath the LCHF is largely devoid of molecular oxygen. We reasoned that the prevalence or absence of DNA repair pathways that cope with oxidative damage would provide insight to the question, are ROS a real and present danger to life at the LCHF?

The guanine oxidation (GO) DNA repair system addresses the most common type of DNA damage caused by reactive oxygen species, the 8-oxo-7,8-dihydroguanine (OG) promutagen (Fig 1) [10]. The OG base differs from guanine by addition of only two atoms, but these change the hydrogen bonding properties so that OG pairs equally well with cytosine and adenine during DNA replication. The resulting OG:A lesions fuel G:C → T:A transversion mutations if not intercepted by the GO DNA repair system [10]. The GO system comprises enzymes encoded by *mutT*, *mutM*, and *mutY*, first discovered through genetic analyses of *Escherichia coli* that demonstrated specific protection from G:C → T:A mutations by these three genes [11–14]. Homologs or functional equivalents of these GO system components are found throughout all three kingdoms of life [15–17], underscoring the importance of the system, yet there are several instances where particular bacteria [18, 19] or eukaryotes [20] make do without one or more of these genes.

Biochemical analyses of gene products have provided a complete mechanistic picture for the GO repair system. MutT hydrolyzes the OG nucleotide triphosphate to sanitize the nucleotide pool, thus limiting incorporation of the promutagen into DNA by DNA polymerase [13, 21]. The enzyme encoded by *mutM*, called formamidopyrimidine-DNA glycosylase (Fpg), locates OG:C base pairs and excises the OG base to initiate base excision repair (BER) [12, 22]. MutY locates OG:A lesions and excises the A base to initiate BER [11, 14]. Fpg and MutY thus act separately on two different intermediates to prevent G:C → T:A mutations. These DNA glycosylases generate abasic (apurinic/apyrimidinic; AP) sites, which are themselves mutagenic if not processed by downstream, general BER enzymes, particularly AP nucleases (e.g. exonuclease III and endonuclease IV), DNA polymerase and DNA ligase [17, 23–25] as shown in Fig 1.

MutY is the final safeguard of the GO system. If left uncorrected, replication of OG:A lesions results in permanent G:C → T:A transversion mutations as demonstrated by *mutY* loss of function mutants [26, 27]. Underperformance of the mammalian homolog, MUTYH, leads to early onset cancer in humans, first discovered for a class of colon cancers now recognized as MUTYH Associated Polyposis [28]. MutY is made up of two domains that both contribute to DNA binding and biochemical functions. The N-terminal catalytic domain shares structural homology with EndoIII and other members of the Helix-hairpin-Helix (HhH) protein superfamily [17]. The C-terminal OG-recognition domain shares structural homology with MutT and other NUDIX hydrolase family members [17, 29]. Functionally important and highly conserved residues define chemical motifs in both domains (Fig 2). These chemical motifs interact with the OG:A lesion and chelate the iron-sulfur cluster cofactor as revealed by x-ray structural analysis (Fig 2) [30–33]. For example, residues in the N-terminal domain establish the catalytic mechanism for adenine excision (Fig 2A and 2B) [32, 34]. Residues found in a beta loop of the C-terminal domain recognize the OG base (Fig 2C), and thus direct adenine removal from

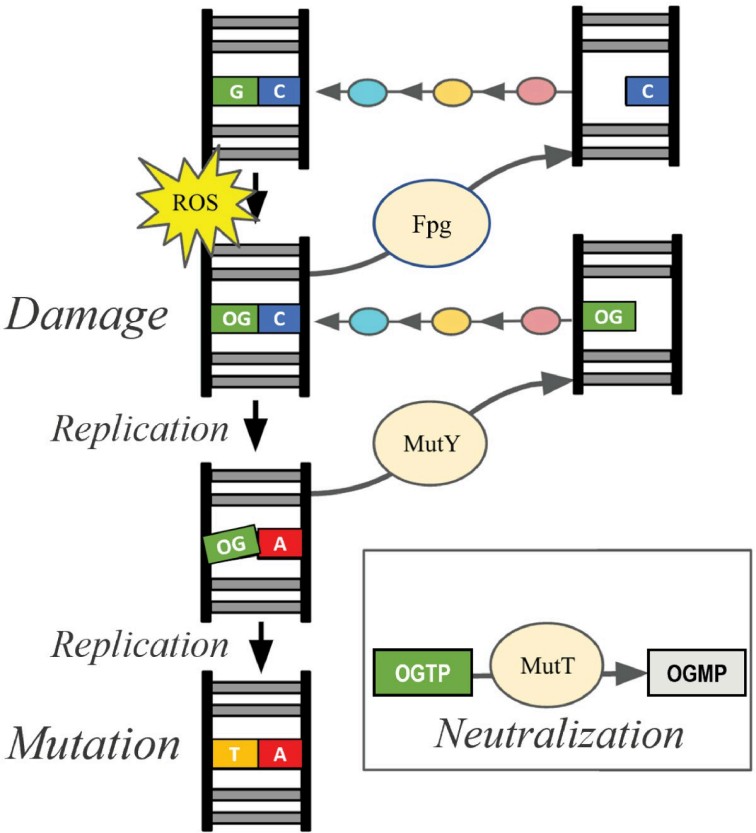

**Fig 1. Overview of the GO system.** The gene products of *mutT*, *mutM*, and *mutY* (tan bubbles) prevent or repair oxidized guanine DNA damage caused by ROS. DNA glycosylases Fpg (encoded by *mutM*) and MutY remove OG from OG:C and A from OG:A, respectively, to create AP sites with no base. Additional enzymes (AP nucleases, pink; DNA polymerase, gold; and DNA ligase, teal) cooperate with the GO pathway, process these AP sites and ultimately restore the GC base pair. MutT neutralizes OG nucleotide triphosphates to prevent incorporation of the OG nucleotide during DNA replication, thereby ensuring that OG found in DNA is on the parent strand, not the daughter strand.

OG:A lesions [33]. Some motifs are shared among other DNA glycosylases, such as the residues that chelate the 4Fe4S iron-sulfur cluster cofactor (Fig 2D) [17]. Chemical motifs particular to MutY, especially the OG-recognition residue Ser 308 (Fig 2C) and supporting residues in the C-terminal domain, are conserved across organisms and are not found in other DNA glycosylases and therefore can be used to identify MutY genes [17].

Our study investigated whether microbes in the anoxic LCHF environment use the GO DNA repair system to mitigate damage caused by reactive oxygen species. It is important to note that not all organisms have an intact GO repair system; examples are missing one, some or all components. MutY in particular was absent frequently in a survey of 699 bacterial genomes [19], and its absence may indicate relaxed evolutionary selection from oxidized guanine damage [18]. We mined for homologous genes within the LCHF microbial community and recombinantly expressed candidate MutY enzymes to characterize function. We found genes encoding GO system components and general base excision repair enzymes at all LCHF sites. MutY homologs from the LCHF suppressed mutations when expressed in *mutY* deficient *E. coli* strains indicating these function similarly to authentic MutY. These Lost City MutY homologs could be assigned confidently to metagenome-assembled genomes (MAG)s, allowing for additional gene inventory analyses that revealed metabolic strategies involving sulfur

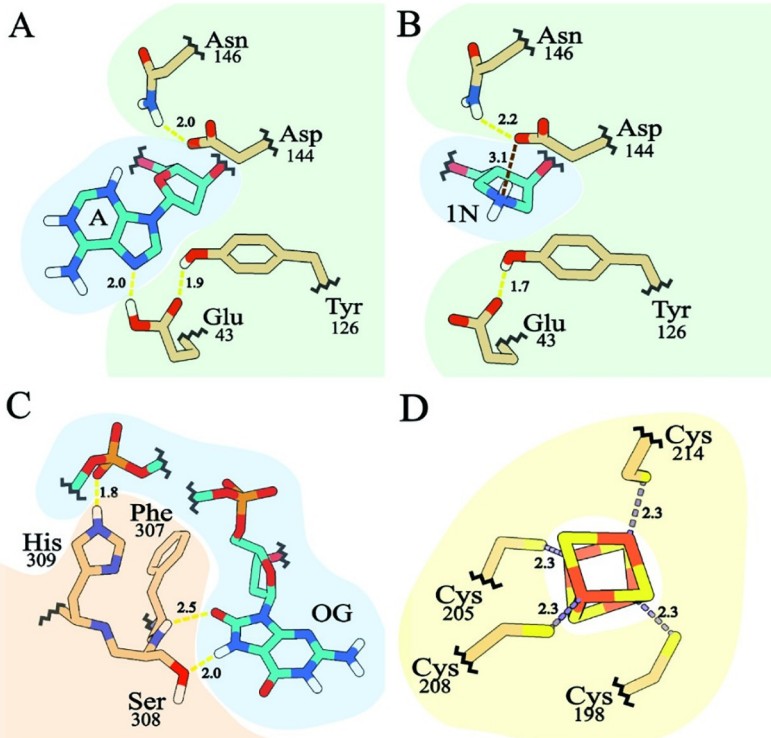

**Fig 2. MutY chemical motifs.** Interactions of MutY residues with DNA and the iron-sulfur cluster cofactor are shown, colored with DNA blue, protein residues tan, and the iron-sulfur cluster yellow/orange. (A) MutY catalytic residues interact with the competitive inhibitor fluoro-A which mimics the substrate adenine. (B) MutY catalytic residues interact with the 1N moiety which mimics the charge and shape of the transition state oxacarbenium ion formed during catalysis of adenine base excision. (C) OG-recognition residues provide hydrogen bonding interactions with the OG base. (D) Four Cys residues chelate the iron-sulfur metal cofactor. All of these interactions are important for MutY activity. Drawn from PDB IDs 3g0q [31] and 6u7t [32].

oxidation and nitrogen reduction. These results have important implications for understanding the repair of oxidative guanine damage in low-oxygen environments, similar to those that existed on a younger Earth, as well as those that may exist on other planets and moons.

## Results

### Identification of the GO DNA repair system in LCHF microbes

To investigate the potential for LCHF microbes to endure DNA damage caused by ROS despite inhabiting a low oxygen environment, we searched for the GO DNA repair system in metagenomes obtained from LCHF hydrothermal fluids [35]. We identified gene homologs for *mutT*, *mutM*, and *mutY*, which constitute the complete GO system (Fig 3). The relative abundances of these GO system gene homologs were similar to that of two other DNA repair enzymes that were also frequently found in LCHF metagenomes. Exonuclease III and endonuclease IV work in conjunction with the GO system and perform general functions necessary for all base excision repair pathways, namely the processing of AP sites [24]. MutY was underrepresented in each of two samples from a chimney named "Marker 3", indicating that this GO system component is not encoded by some of the LCHF residents (Fig 3, magenta).

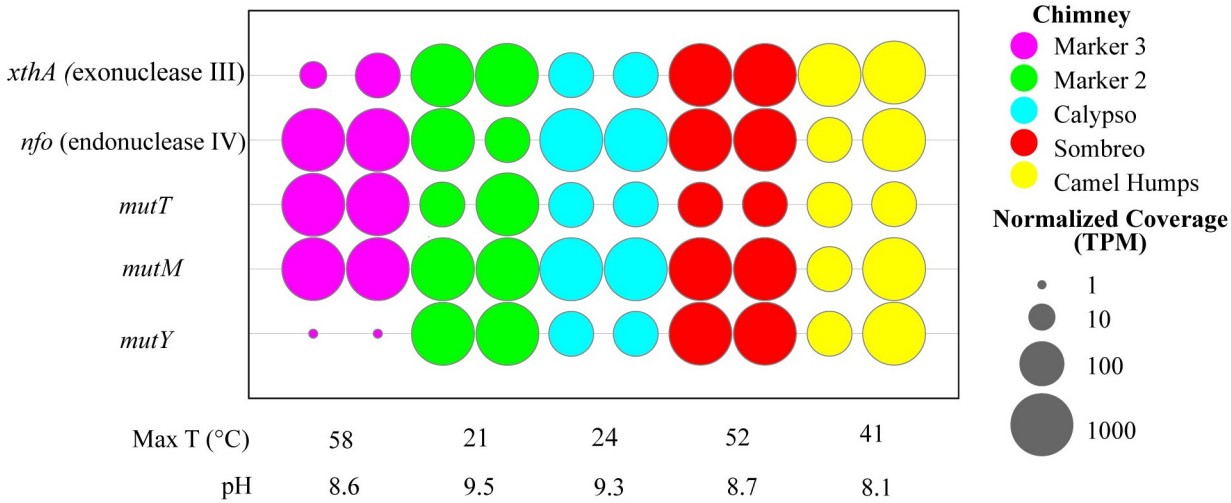

**Fig 3. Abundance of GO system gene homologs.** Listed on the vertical axis are genes encoding DNA repair enzymes. Genes *xthA* and *nfo* are generally necessary for DNA repair involving base excision repair in bacteria, including the particular GO system investigated here. Together, *mutT*, *mutM* and *mutY* constitute the GO system that deals specifically with oxidized guanine. Across the horizontal axis are the various LCHF sites, coded by color, from which samples were collected in duplicate, along with the reported temperature and pH. The normalized coverage of each gene is reported as a proportional unit suitable for cross-sample comparisons, the transcripts/fragments per million (TPM).

## Metagenomic mining for MutY genes

Having determined that GO system gene homologs are abundant at the LCHF, we focused our efforts on the final safeguard of the pathway, MutY. A BLASTP search against the LCHF metagenome with query MutY sequences from *Geobacillus stearothermophilus* (*Gs* MutY) and *E. coli* (*Ec* MutY) preliminarily identified 649 putative MutY candidates on the basis of sequence identity, excluding hits with less than a 30% sequence identity cut-off or E-values exceeding 1E-5 (Fig 4A). Structure-guided alignments of these preliminary hits were examined for presence and absence of MutY-defining chemical motifs. We paid particular attention to the chemical motif associated with OG recognition as these residues in the C-terminal domain establish OG:A specificity, which is the hallmark of MutY [29, 33, 36]. This approach authenticated 160 LCHF MutYs (Fig 4B). Four representative LCHF MutYs were selected for further analyses described below (red branches in Fig 4A and 4B).

Fig 5 highlights conservation and diversity for the MutY-defining chemical motifs found in the 160 LCHF MutYs. All of the LCHF MutYs retain the chemical motif to coordinate the iron-sulfur cluster cofactor comprising four invariant Cys residues (C, yellow in Fig 5), a feature that is also found in other HhH family members such as EndoIII [16], but which is absent for some "clusterless" MutYs [37]. Other invariant and highly conserved motifs make critical interaction with the DNA and provide key catalytic functions for adenine base excision, explaining the high degree of sequence conservation at these positions. For example, all LCHF MutYs use a Glu residue which provides acid base catalysis for the mechanism (first E, green in Fig 5). Also, all LCHF MutYs use a Gln (first Q, red in Fig 5) and a Tyr (first Y, red in Fig 5), to wedge between base pairs and thereby distort the DNA for access to the adenine as seen in x-ray crystal structures of *Gs* MutY [30, 31]. Structures of *Gs* MutY interacting with a transition state analog revealed close contact with Tyr126 (second Y, green in Fig 5), Asp144 (D, green in Fig 5), and supported by Asn146 (N, green in Fig 5) indicating these chemical motifs stabilize the transition state during catalysis [32–34]. For the LCHF MutYs, the Asn residue is

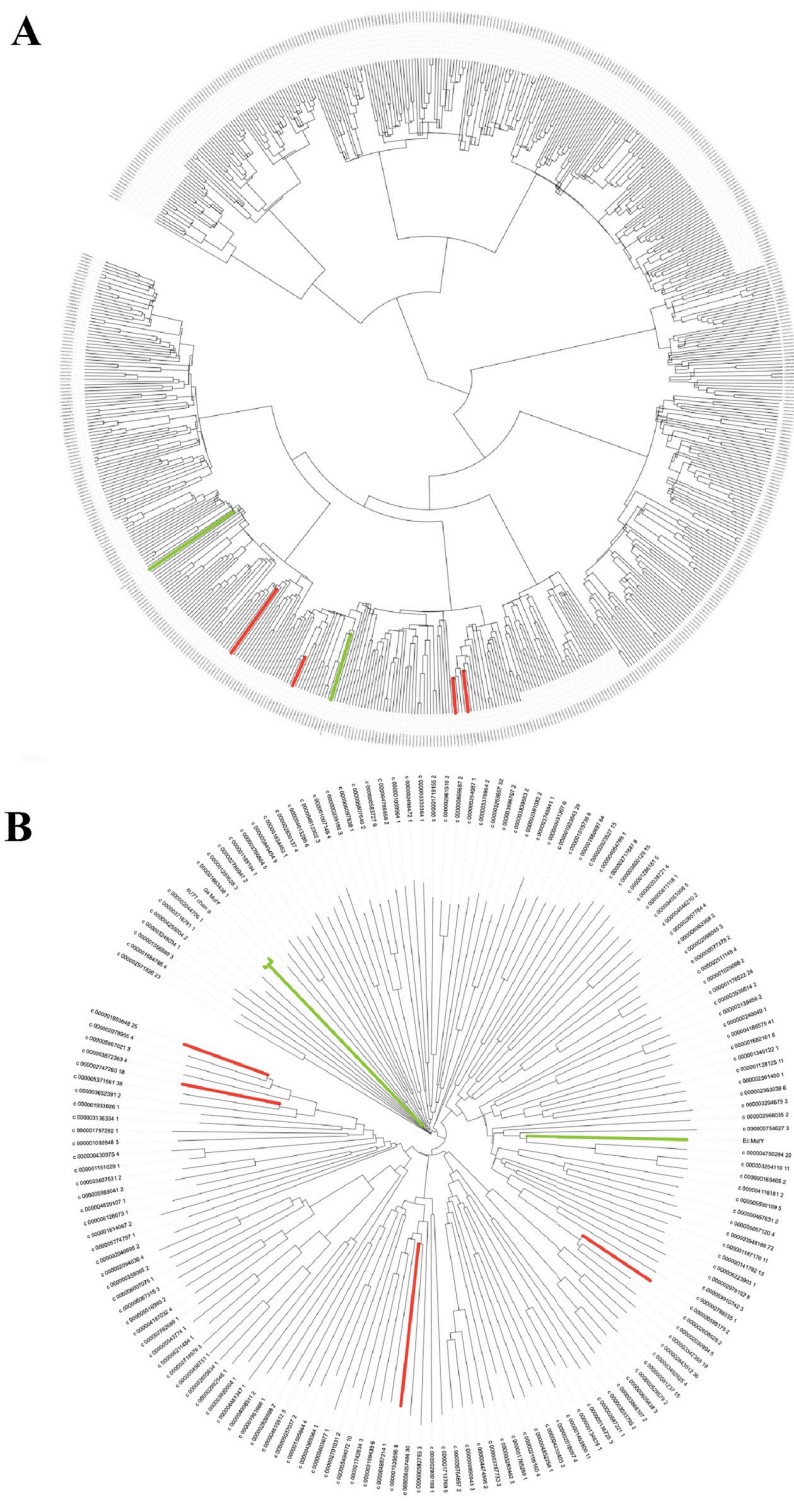

**Fig 4. Phylogeny for LCHF MutYs.** (A) 649 sequences were identified as LCHF MutY candidates due to sequence similarity to existing MutYs (green branches) and aligned to reconstruct evolutionary relationships. (B) A subset of 160 members contained all necessary MutY-defining chemical motifs. Alignment of these authenticated LCHF MutYs revealed varying evolutionary distances from familiar MutYs and provided a basis for selecting four representative members (red branches).

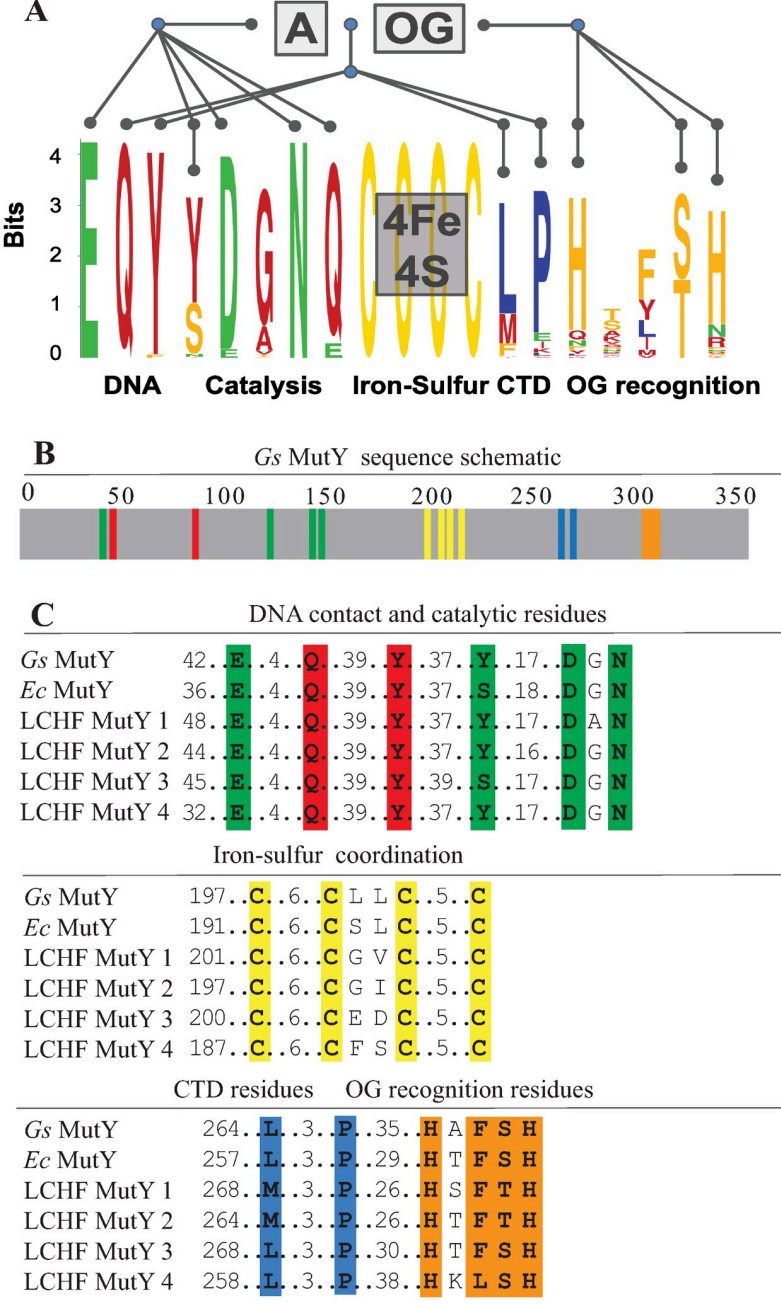

**Fig 5. LCHF MutY chemical motifs.** (A) Conservation and diversity of MutY-defining chemical motifs are depicted with a sequence logo for the 160 LCHF MutYs. These motifs are associated with biochemical functions including DNA binding, enzyme catalysis, attachment of the iron-sulfur cofactor, and recognition of the damaged OG base. See S1 Dataset for a complete alignment and S1 Table for a percent identity matrix for the representative LCHF MutYs. (B) Chemical motifs located in MutY as shown with color-coded positions for *Gs* MutY. (C) Alignment for select chemical motifs highlights conservation among the four representative LCHF MutYs, *Ec* MutY and *Gs* MutY. Sequence logo generated by Weblogo [38, 39].

invariant, the catalytic Asp is nearly invariant, replaced by chemically similar Glu for five LCHF MutYs, and the catalytic Tyr residue is often replaced by Ser and sometimes by Thr or Asn. The residue found between the catalytic Asp and Asn is always a small residue, most often Gly (G, red in Fig 5) but sometimes Ala, Val or Thr.

By contrast with these numerous and highly conserved motifs for the N-terminal domain, fewer motifs with greater sequence divergence were found in the C-terminal domain. The MutY ancestor is thought to have resulted from a gene fusion event that attached a MutT-like domain to the C-terminus of a general adenine glycosylase enzyme, and the C-terminal domain of modern *Ec* MutY confers OG:A specificity [29]. X-ray crystal structures of *Gs* MutY interacting with OG:A and with G:A highlighted conformational difference for a Ser residue in the C-terminal domain (Ser308 in *Gs* MutY), and mutational analysis showed this Ser residue and its close neighbors (Phe307 and His309 in *Gs* MutY) establish OG specificity [33]. Informed by these insights from structure, we eliminated LCHF MutY candidates that lacked a C-terminal domain and its OG-recognition motif. Alignment of the 160 LCHF MutYs that passed this test showed that a second His residue is also well conserved within the H-x-FSH sequence motif (Fig 5C). As is evident from the alignment, there are many variations with residues replaced by close analogs at each position. Ser, which makes the key contacts with N7 and O8 of OG and no contact with G, is often replaced by Thr, which can make the same hydrogen bond interactions. Likewise the two His residues are each often replaced by polar residues (e.g. Gln, Asn, Arg or Lys) that can also hydrogen bond to the DNA phosphate backbone as observed for His305 and His309 in *Gs* MutY.

Two other positions with high conservation were revealed for the C-terminal domain in this analysis of the 160 LCHF MutYs. These define a L-xxx-P motif. These residues are replaced by other residues with comparable chemical properties. The Leu position is often another hydrophobic residue such as Met or Phe, and the Pro position is most frequently replaced by Glu, a residue that can present aliphatic methylene groups and thus resemble Pro if the polar group hydrogen bonds with the peptide amide. In the structure of *Gs* MutY, the Pro269 nucleates a hydrophobic core for the C-terminal domain. The Leu265 makes a strong VDW contact with Tyr89 in the N-terminal domain to support stacking of Tyr88 between bases of the DNA, a molecular contact that suggests communication between the OG-recognition domain and the catalytic domain. Other evolutionary analyses have highlighted the motifs important for DNA contacts, catalysis and OG recognition [17], but the L-xxx-P motif has not been identified previously.

Four representative LCHF MutYs were selected for further analyses. S1 Table reports the percent identity among these representative LCHF MutYs and the well-studied MutYs from *E. coli* and *G. stearothermophilus*. LCHF MutY 1 and LCHF MutY 2 are most closely related with 65% identity which is almost twice the average in this group. LCHF MutY 3 is most closely related to *Ec* MutY with 48% identity. We examined the representative LCHF MutYs for physical properties as inferred from sequences. Table 1 reports these physical properties including predicted protein size, isoelectric point (pI), and stability (Tm). Generally the physical characteristics measured for LCHF MutY representatives were comparable to each other and to predicted properties of *Ec* MutY and *Gs* MutY. The predicted Tm for LCHF MutY 3 was above 65°C, distinguishing it as the most stable enzyme (Table 1), which may reflect adaptation to a high temperature environment. The isoelectric point predicted for each of the LCHF MutY representatives is 3 pH units above the pI predicted for *Gs* MutY and between 0.1–0.5 pH units above the pI predicted for *Ec* MutY, indicating that more numerous positively charged residues have been recruited, possibly as an adaptation to the LCHF environment.

Table 1. Physical protein properties.

| MutY | Length (residues) | MW (kDa) | pI | Predicted Tm (˚C) (N-domain; C-domain) |
|---|---|---|---|---|
| *Gs* MutY | 372 | 41.8 | 5.3 | 55–65 (55–65; < 55) |
| *Ec* MutY | 355 | 39.1 | 8.6 | < 55 (< 55; 55–65) |
| LCHF MutY 1 | 358 | 39.0 | 9.1 | 55–65 (55–65; < 55) |
| LCHF MutY 2 | 352 | 38.3 | 8.8 | < 55 (55–65; < 55) |
| LCHF MutY 3 | 370 | 42.0 | 8.7 | > 65 (>65; 55–65) |
| LCHF MutY 4 | 376 | 44.0 | 9.0 | 55–65 (55–65; 55–65) |

## Identification of LCHF MutY organisms, gene neighbors, environmental conditions, and metabolic strategies

Our next objective was to identify the organisms from which these LCHF MutY enzymes originated. Each of the four representative LCHF MutY sequences were derived from contiguous DNA sequences (contigs) belonging to a MAG representing a LCHF microbe. The taxonomic classification of these MAGs indicated that LCHF MutY 1 originated from a species of *Marinosulfonomonas*, LCHF MutY 2 from the family *Rhodobacteraceae*, LCHF MutY 3 from the family *Thiotrichaceae*, and LCHF MutY 4 from the family *Flavobacteriaceae* (Fig 6). The taxonomic classification of each contig was consistent with the classification of the MAG to which it belonged, supporting the idea that the MutY gene is a long term resident and not a recent arrival through phage infection or some other horizontal gene transfer mechanism. For the remainder of this work we will refer to the MutY-encoding organisms by the lowest-level classification that was determined for each LCHF MutY (e.g. LCHF MutY 3 will now be referred to as *Thiotrichaceae* MutY).

The inclusion of MutY contigs in MAGs provided an opportunity to examine gene neighbors for the representative LCHF MutYs. The GO repair genes are located at distant loci in *E. coli* [12], and belong to separate operons [40]. However, MutY is the immediate 5'-neighbor to YggX within gammaproteobacteria [40], and homologs of YggX are present outside this lineage, occasionally nearby to MutY (e.g. *Bacillus subtilis*). As gene neighbors, MutY and YggX are part of a SoxRS regulated operon in *E. coli* [40, 41]. YggX provides oxidative stress protection and iron transport function with a critical Cys residue close to the N-terminus of this small protein [42, 43]. A protein matching these features is encoded by a gene partly overlapping with and the nearest 3' neighbor to *Thiotrichaceae* MutY (see S1 Fig).

To reveal the environmental conditions of these MutY-encoding organisms, we analyzed the sequence coverage of each LCHF MutY contig at each of the sampling sites. *Marinosulfonomonas* MutY, *Thiotrichaceae* MutY, and *Flavobacteriaceae* MutY were identified at all sampling sites, ranging from 21˚C to 58˚C and pH 8.1 to 9.5. *Rhodobacteraceae* MutY was present at all sampling sites excluding Marker 3 and was therefore found in temperatures ranging from 24˚C to 52˚C and pH 8.1 to 9.5.

We further investigated the metabolic strategies utilized by MutY-encoding microbes by examining the inventory of predicted protein functions in each MAG (Table 2, see also S2 Table). Each LCHF MutY-containing MAG possessed at least two forms of cytochrome oxidase, with the exception of the *Flavobacteriaceae* MAG. The *Flavobacteriaceae* MAG is only

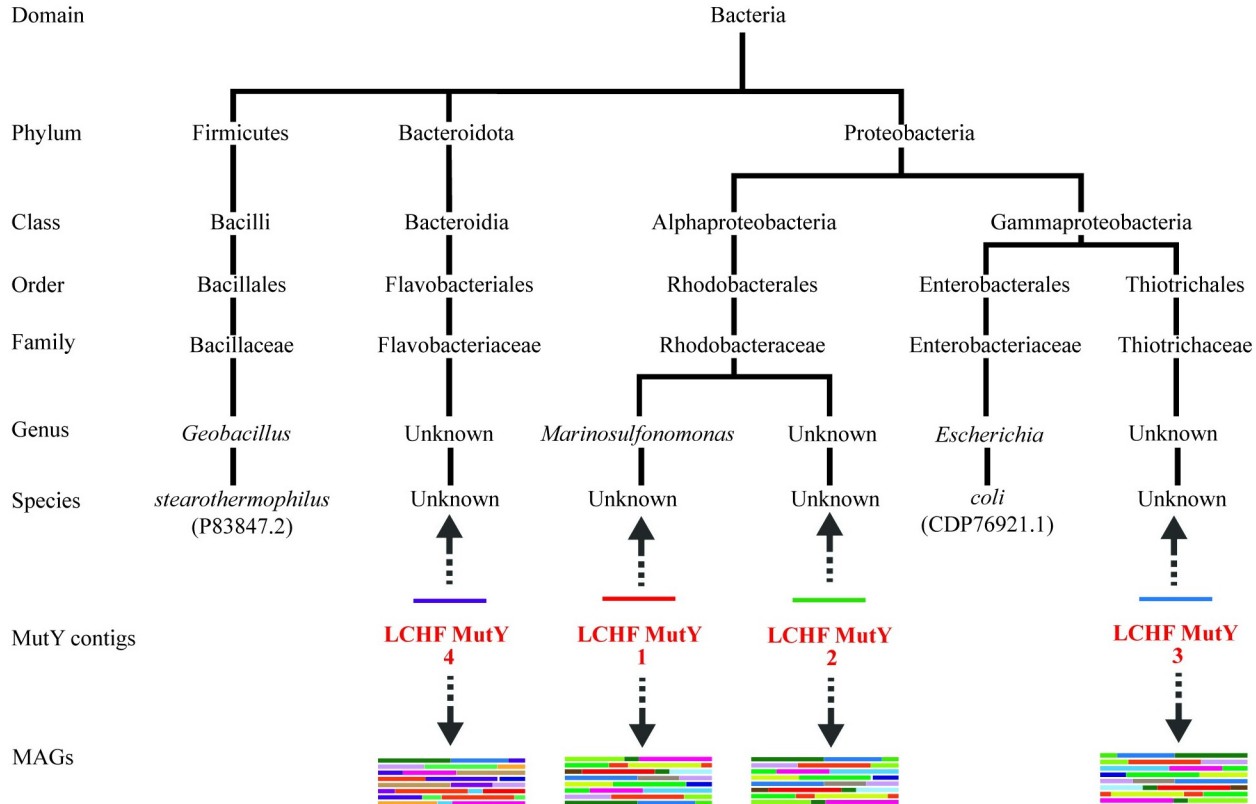

**Fig 6. Taxonomic classification.** LCHF MutY-encoding contigs were found in several branches of bacteria. The classification places these in relation to MutY of *G. stearothermophilus* and *E. coli* (accession IDs included). LCHF MutYs were mapped to their respective microbes by two methods indicated by the two arrows (see text for details).

44% complete, however, so no strong conclusions can be made regarding the absence of genes. Cytochrome oxidases commonly provide sources of free radicals and are essential to aerobic metabolism. Predicted proteins indicative of dissimilatory nitrate and nitrite reduction were found in the *Marinosulfonomonas*, *Rhodobacteraceae*, and *Thiotrichaceae* MAGs, suggesting that these organisms may be capable of using nitrate or nitrite as alternative electron acceptors when oxygen is not available. Furthermore, the *Marinosulfonomonas* and *Rhodobacteraceae* MAGs include predicted protein functions associated with the oxidation of reduced sulfur compounds, though it is important to note that the directionality of these reactions cannot be fully determined from bioinformatics alone. These patterns speak to the potential origins of oxidants within the MutY-encoding organisms as discussed below.

## Predicted protein structures and virtual docking experiments

To assess the likelihood of the LCHF MutY sequences folding into enzymes capable of activity on the OG:A substrate, protein structures were predicted using *Colabfold* [44] (Fig 7). These predicted structures were associated with high confidence as indicated by pLDDT scores and PAE profiles (Table 3 and S2 Fig). Superpositions revealed that the predicted structures for the LCHF MutYs are each highly comparable with the experimentally determined structure for *Gs* MutY as indicated by visual inspection (Fig 7) and by low, pairwise root mean square deviation (RMSD) values (Table 3). The whole protein superpositions were dominated by the larger,

**Table 2. Metabolic genes identified in LCHF MutY organisms.**

| Metabolism | Gene | KEGG id | *Marinosulfonomonas* [a] | | *Rhodobac-teraceae* | *Thiotrich-aceae* | *Flavobac-teriaceae* |
|---|---|---|---|---|---|---|---|
| | | | MAG 1 | MAG 2 | | | |
| Cytochrome Oxidases | UQCRFS1 | K00411 | X | X | X | X | |
| | coxA | K02274 | | | X | X | |
| | ccoN | K00404 | X | X | X | | |
| | cydA | K00425 | | X | | | |
| | cyoB | K02298 | | | X | | |
| Sulfur Oxidation | soxA | K17222 | X | X | X | | |
| | soxX | K17223 | X | X | X | | |
| | soxB | K17224 | X | X | X | | |
| | soxC | K17225 | X | X | X | | |
| | soxY | K17226 | X | | X | | |
| | soxZ | K17227 | X | | X | | |
| Nitrogen Reduction | narG | K00370 | X | X | X | | |
| | narH | K00371 | X | X | X | | |
| | nirB | K00362 | | X | X | X | |
| | nirD | K00363 | | X | | X | |
| | nirK | K00368 | | | X | | |
| | norB | K04561 | | | X | | |
| | norC | K02305 | | | X | | |
| | nosZ | K00376 | | | X | | |
| MAG Completeness (%) [b] | | | 88.4 | 88.2 | 93.7 | 66.1 | 44.3 |
| MAG Contamination (%) [b] | | | 16.4 | 0.6 | 1.4 | 11.8 | 1.6 |

[a] *Marinosulfonomonas* MutY belongs to two separate MAGs and genes for each are reported separately.

[b] Completeness and contamination scores generated by *CheckM* v1.0.5 as described in Brazelton *et al.* 2022 [35].

more structurally conserved N-terminal domain. Breaking the analysis into two separate domains showed that the C-terminal domain, although more plastic, retained core structural features that could be superimposed. The MutY-defining chemical motifs are positioned in locations similar to those seen for the *Gs* MutY reference structure, providing evidence these LCHF MutY enzymes are capable of recognizing OG:A lesions and excising the adenine base. Concisely, the LCHF MutY structure predictions resemble a functional MutY enzyme from a thermophilic bacterium.

We performed virtual docking experiments to examine the potential for molecular interaction with adenosine and OG ligands. MutY scans DNA looking for the OG:A base pair by sensing the major-groove disposition of the exocyclic amine of the OG base in its *syn* conformation [51, 52]. After this initial encounter, the enzyme bends the DNA, flips the adenine base from the DNA double helix into the active site pocket, and positions OG in its *anti* conformation as seen in structures of the enzyme complexed to DNA [30, 31]. Thus, multiple conformations and orientations for the OG and adenosine ligands were anticipated. The search volume for the adenosine ligand was centered on the active site in the NTD, and the search volume for the OG ligand was defined by the OG-recognition motif found in the CTD. Representative outcomes obtained with *Autodock VINA* are shown in Fig 7, and the corresponding binding affinities are reported in Table 3 and S3 Table. As anticipated the precise orientation and position for these docked ligands varied, and none exactly match the disposition of the adenine or OG base as presented in the context of double stranded DNA (Fig 7A). Nevertheless, binding

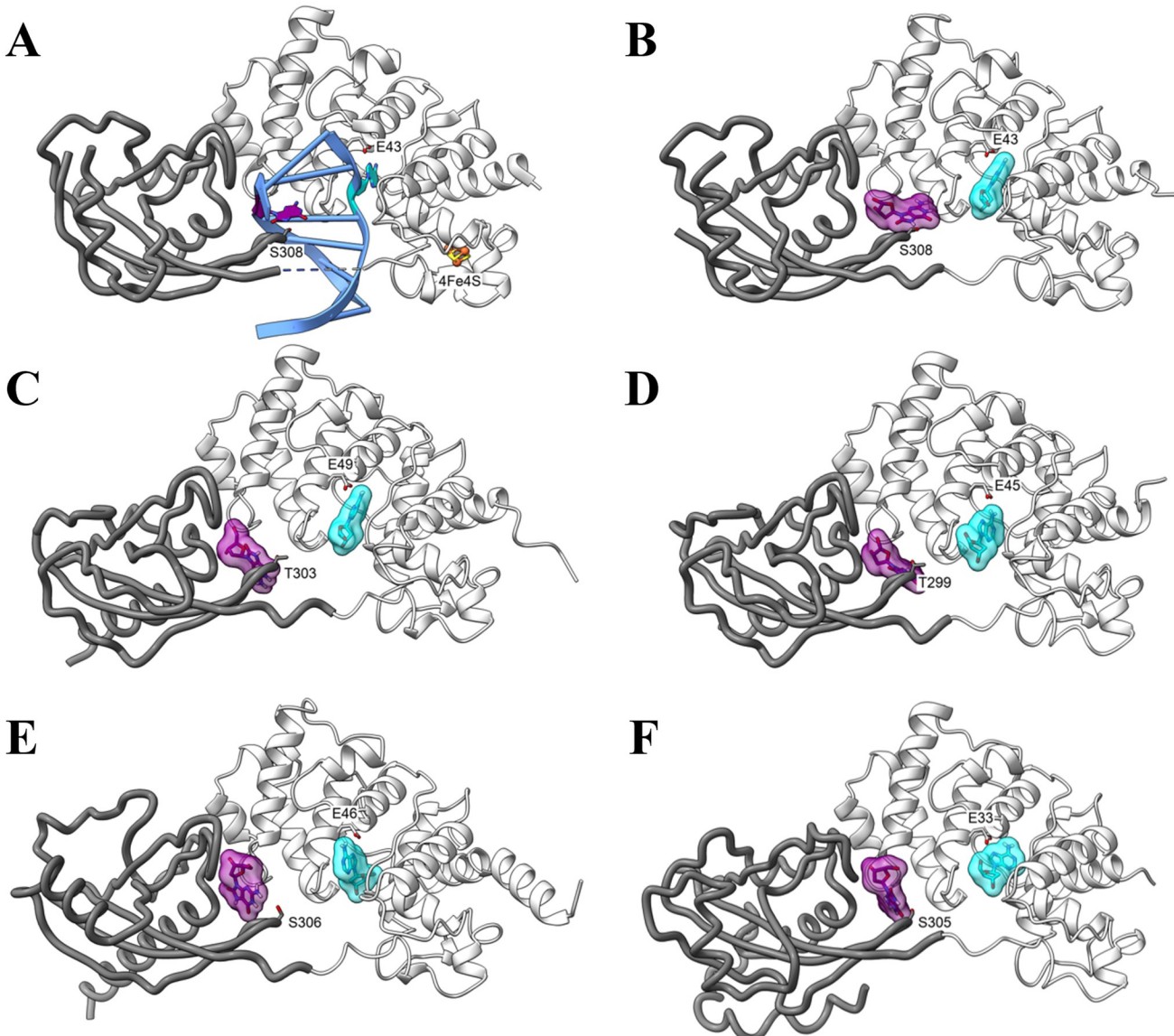

**Fig 7. Structure predictions and virtual docking of MutY ligands.** (A) The x-ray crystal structure of *Gs* MutY (white ribbon NTD; grey CTD; PDB ID 3g0q) in complex with DNA (blue) highlights the positions of the adenine nucleotide (cyan) and OG (purple). (B-F) Virtual docking of ligands. (B) Adenosine and OG were separately docked to identify binding surfaces for these ligands in the structure of *Gs* MutY (PDB ID 6u7t)), which served as a positive control. (C-F) Docking outcomes for the four representative LCHF MutYs: *Marinosulfonomonas* MutY (C); *Rhodobacteraceae* MutY (D); *Thiotrichaceae* MutY (E); and *Flavobacteriaceae* MutY (F).

affinities for the ligand-LCHF MutY complexes ranged from -6.8 to -8.0 kcal/mol, indicating favorable interactions were attainable and similar to the binding affinities measured for *Gs* MutY by the same virtual docking method.

## Molecular dynamics simulations

Virtual docking is fast and computationally economical but largely ignores motion and solvent. The reliability of docking improves when complemented with molecular dynamic (MD) simulation [53, 54]. To further assess stability and dynamic properties of LCHF MutY-ligand

**Table 3. Molecular modeling for LCHF MutYs.**

| MutY Model | pLDDT [d] | RMSD (Å) [a] (residues) | | Affinity *VINA* (kcal/ mol) [b] | | Energy *Amber* [c] (kJ/mol) | |
| --- | --- | --- | --- | --- | --- | --- | --- |
| | | NTD | CTD | A | OG | A | OG |
| *Gs* MutY (6u7t) [e] | NA | 0.39 (216) | 0.54 (116) | -7.3 | -7.7 | -213 (±38) | -168 (±34) |
| *Marinosulfonomonas* | 93 (±9) | 0.87 (198) | 0.90 (64) | -6.8 | -7.5 | -80 (±37) [c] | -194 (±44) |
| *Rhodobacteraceae* | 94 (±6) | 0.85 (199) | 0.98 (51) | -6.8 | -7.7 | -170 (±37) | -110 (±54) [c] |
| *Thiotrichaceae* | 92 (±12) | 1.0 (195) | 1.1 (77) | -7.0 | -8.0 | -226 (±39) | -205 (±52) |
| *Flavobacteriaceae* | 92 (±14) | 0.84 (199) | 1.2 (44) | -7.2 | -8.0 | -214 (±34) | -197 (±30) |

[a] Root mean square deviation for the superposition of predicted structures with *Gs* MutY (PDB ID 3g0q) was calculated separately for N-terminal domain (NTD) and C-terminal domain (CTD) by *ChimeraX* [45].

[b] Binding affinity for the best outcome from docking adenosine to the enzyme active site and OG to the OG-recognition site as calculated by *Autodock VINA* [46, 47].

[c] Energy for short-range Coulombic and Lennard-Jones interactions with the ligand as computed by *GROMACS* [48], with the *Amber*99SB and GAFF force fields [49, 50]. Energies were averaged over the 100-ns simulation or the time window of the complex, 0–26 ns for *Mainosulfonomonas* interaction with A and 0–47.5 ns for *Rhodobacteraceae* interaction with OG. Uncertainty is the sample standard deviation.

[d] Local distance difference test metric to assess confidence for structures predicted by *Colabfold* [44] averaged over all residues. Uncertainty is the sample standard deviation.

[e] Reference superposition values provided by comparing a second structure of *Gs* MutY in complex with its transition state analog (PDB ID 6u7t)

complexes derived by docking, we applied MD simulations with the *Amber* force field [49, 50], as implemented with *GROMACS* [48]. Each protein-ligand complex was solvated in water, charges were balanced with counterions, and the system was equilibrated in preparation for a 100-ns MD simulation. S3 Fig and S1 Movie summarize the resulting trajectories in terms of interaction energy, distance, and structure over time. We focused on mechanistically relevant interactions by tracking distances from the base moiety to the catalytic Glu residue for adenosine complexes, and distances to OG-recognition Ser and Thr residues for OG complexes. MD trajectories for the *Gs* MutY-ligand complexes (S3A and S3B Fig, S2 and S3 Movies) provided a basis of comparison for the LCHF MutY-ligand complexes.

MD analysis revealed dynamic and, in some cases, unstable complexes. Relative instability likely reflects the free nature of the ligands, which normally would be presented as part of DNA. As will become evident in later sections, complex instability detected by MD simulation correlates positively with biological activity under mesophile conditions. Even so, many of the MutY-ligand complexes persisted for the entire 100-ns simulation, characterized by favorable binding affinity, extracted as the sum of local Lennard-Jones and Coulombic interactions (Table 3). While all ligands were mobile, the MD outcomes separated into two groups distinguished by the degree of ligand movement and persistence of the complex. In the first group the adenosine and OG ligands remained close to the original binding sites for at least 90 ns if not the entire 100-ns MD simulation. This first group with persistently engaged ligands included the complexes with *Gs* MutY (S3A and S3B Fig, S2 and S3 Movies), *Thiotrichaceae* MutY (S3G and S3H Fig, S8 and S9 Movies) and *Flavobacteriaceae* MutY (S3I and S3J Fig, S10 and S11 Movies).

For example, adenosine remains bound to the active site of *Thiotrichaceae* MutY for the entire 100-ns MD simulation. Catalytic Glu46 made contact with N7 of adenosine via a bridging solvent molecule, with this mechanistically relevant interaction observable for the first 11 ns (S3G Fig and S8 Movie). Water-mediated interaction of the catalytic Glu and N7 was also observed for *Gs* MutY (S3A Fig and S2 Movie), and is comparable to water-bridging interactions described previously in MD simulations of *Gs* MutY complexed to double stranded DNA

by others [55]. Indeed, such water-mediated interaction was first observed in the crystal structure of *Gs* MutY complexed to substrate DNA [30]. Thus, our MD analysis captures interactions of functional importance despite lacking a full treatment of DNA.

Similar to observations for adenosine, OG remained bound at the interface of the NTD and CTD in its complex with *Thiotrichaceae* MutY and with *Flavobacteriaceae* MutY, despite notable interdomain hinge motion and flexibility in the CTD. For *Thiotrichaceae* MutY, Ser306 engaged the OG ligand via hydrogen bonds to N1, N2 and O6 of the Watson-Crick-Franklin face during the first 39 ns (S3H Fig and S9 Movie). Interactions with the Watson-Crick-Franklin face of OG, especially with N2 presented in the major groove, are known to facilitate initial recognition of the OG lesion [51, 52]. Crystal structures feature the corresponding Ser of *Gs* MutY hydrogen bonded with N7 and O8 of OG [30–32], and similar contacts between Ser305 and N7, O8 and N6 of the Hoogsteen face are observed during the first 13 ns for *Flavobacteriaceae* MutY complexed to OG (S3J Fig and S11 Movie).

By contrast with these persistently engaged ligands observed in the first group, ligands in the second group disengaged and departed from the original binding site and found new sites within the first 10 ns, as observed for complexes with *Marinosulfonomonas* MutY (S3C and S3D Fig, S4 and S5 Movies) and *Rhodobacteraceae* MutY (S3E and S3F Fig, S6 and S7 Movies). During the *Marinosulfonomonas* MutY simulation, adenosine slipped out of the active site pocket within 1 ns, remained near the active site entrance until 6.4 ns, when it exited completely and engaged with several different sites on the protein surface (S3C Fig and S4 Movie). The situation was comparable for adenosine complexed to *Rhodobacteraceae* MutY, but the ligand found a resting place after departing the active site pocket (S3E Fig and S5 Movie), wedged into a groove with residues Gly126 and Tyr128 on one side and Gln49 and Arg93 on the other side. This alternate adenosine binding site for *Rhodobacteraceae* MutY is adjacent and partially overlapping with the exosite observed for cytosine in the complex of *Gs* MutY with its OG:C anti-substrate [56]. Departure of the base from the active site as observed in our MD simulations was anticipated since crystal structures of MutY in complex with enzyme-generated abasic site (AP) product show no electron density for the base moiety [34], implying that the free base has an escape route.

Binding site departure was also observed for the OG ligand, which disengaged from the CTD of *Marinosulfonomonas* MutY and found new binding sites on the surface of the NTD, as the two domains hinged away from each other (S3D Fig and S5 Movie). At the outset, OG bound to *Rhodobacteraceae* MutY with OG-specific hydrogen bonds connecting Thr299, N7 and O8 atoms (S3F Fig and S7 Movie), very comparable to hydrogen bonds seen in crystal structures of *Gs* MutY bound to DNA with the OG lesion [30–32]. However, the FTH loop of *Rhodobacteraceae* MutY pulled away early in the MD simulation at 4.4 ns, thereby breaking these hydrogen bonds. The OG ligand subsequently adopted several novel poses at sites on the NTD and alternatively on the CTD before dissociating completely by 48 ns (S3F Fig and S7 Movie).

In summary, MD simulations differentiated the LCHF MutYs into two groups based on conformational flexibility and ligand persistence. Ligand persistence was also observed for the complexes with the x-ray crystal structure of *Gs* MutY. Kinetically unstable ligand complexes observed for *Marinosulfonomonas* and *Rhodobacteraceae* MutYs prompted further *in vivo* validation to address the open question, which enzyme, if any, could support biological function?

## Testing mutation suppression activity of LCHF MutY enzymes by recombinant expression

The *in silico* experiments provided strong evidence that the LCHF MutYs are structurally comparable to authentic MutY enzymes, with affinity for OG:A lesions, albeit with kinetic

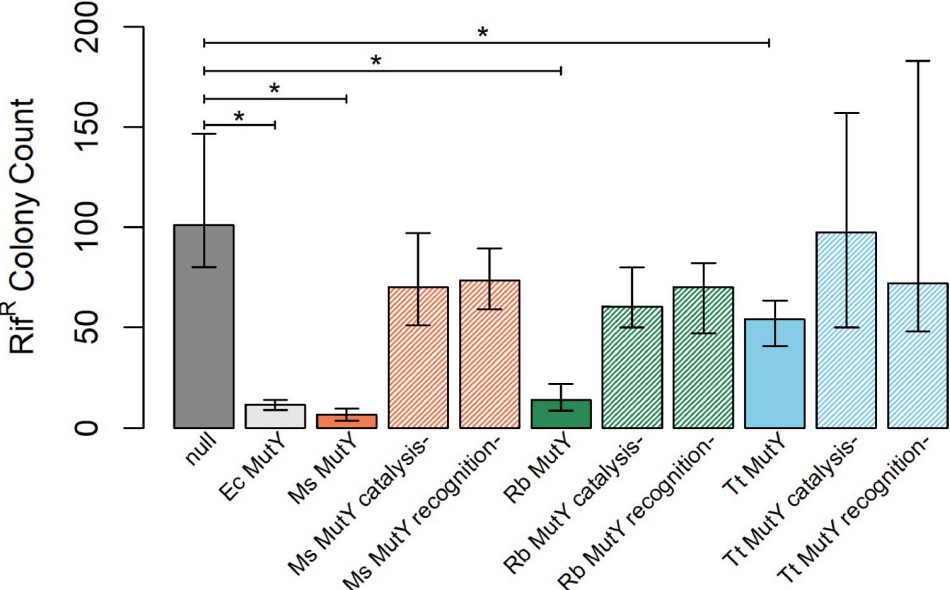

**Fig 8. Functional analysis.** Bars represent median Rif$^R$ colony counts for *E. coli* cultures expressing MutY, MutY variants, or no MutY (*null*) from a plasmid DNA. Error bars represent 95% confidence intervals as determined by bootstrap sampling (see S4 Table for values). *Marinosulfonomonas* (*Ms*) MutY, *Rhodobacteraceae* (*Rb*) MutY, and *Thiotrichaceae* (*Tt*) MutY each suppressed mutations as evidenced by non-overlap of Rif$^R$ confidence intervals compared to *null* cultures. Altered versions of each LCHF MutY tested the importance of residues for catalysis and OG-recognition. Designated *catalysis-* and *recognition-* along the X-axis, these alterations severely impacted mutation suppression function indicating the LCHF MutYs share mechanistic features with the extensively studied enzymes *Ec* MutY and *Gs* MutY.

instability in notable cases, suggesting these may function to prevent mutations. To demonstrate biological function directly, we recombinantly expressed the genes in *E. coli* and measured mutation suppression activity *in vivo*. Three of the representative LCHF MutYs were successfully cloned into the pKK223 expression plasmid as verified by Sanger sequencing. The *Flavobacteriaceae* MutY appeared to be toxic to *E. coli* as only mutant versions of the gene were obtained from multiple cloning attempts, a situation that is reminiscent of *Gs* MutY, which is also apparently toxic to *E. coli* and could not be cloned into pKK223 [33].

To test the mutation suppression activity of *Marinosulfonomonas*, *Rhodobacteraceae*, *and Thiotrichaceae* MutYs, we measured mutation rates with a rifampicin resistance assay [57]. Several, independent, single-point mutations within the gene encoding RNA polymerase beta-subunit (*rpoB*) confer antibiotic resistance [58, 59]. Thus, spontaneous Rif$^R$ mutants arising in overnight cultures can be counted by the colonies that emerge on rifampicin containing plates. Cultures expressing functional MutY delivered by plasmid DNA transformation have low Rif$^R$ frequency compared to the high Rif$^R$ frequency characterizing the reporter strain that lacks *mutY* and *mutM* genes (see Materials and methods).

Cultures with an empty plasmid (*null*) and cultures with a plasmid encoding *Ec* MutY showed significant differences in the frequency of Rif$^R$ mutants, with median values of 101 and 12, respectively, indicating the assay was fit for use (significance determined by non-overlap of median 95% confidence intervals) (Fig 8 and S4 Table). *Marinosulfonomonas* MutY, *Rhodobacteraceae* MutY, and *Thiotrichaceae* MutY each demonstrated significant mutation suppression activity when compared to the *null*. Indeed, *Rhodobacteraceae* MutY showed mutation suppression performance equivalent to that measured for *Ec* MutY, and *Marinosulfonomonas*

MutY was apparently better at suppressing mutations than *Ec* MutY (Fig 8 and S4 Table), a remarkable outcome given the evolutionary time separating these species. Note that these LCHF MutYs with high mutation suppression function formed unstable complexes as revealed by MD simulation. *Thiotrichaceae* MutY showed partial function. Cultures expressing *Thiotrichaceae* MutY suppressed Rif$^R$ mutants to 50% of the rate observed for *null* cultures, but allowed a mutation rate about 4-fold greater than that measured for cultures expressing *Ec* MutY (Fig 8 and S4 Table).

To investigate the biochemical mechanism employed by LCHF MutY enzymes, we altered residues essential for OG:A recognition and catalysis, then repeated the mutation suppression assay. Two mutants of each LCHF MutY were constructed through site-directed substitution of residues. One set of substitutions was designed to disable the OG-recognition motif by replacing F(S/T)H residues (Figs 2 and 5) with alanine residues (designated *recognition-*); the other set of substitutions was designed to disable catalysis by replacing the active site Asp and Glu residues with structurally similar, but chemically inert Asn and Gln residues (*catalysis-*). For all three LCHF MutYs, these targeted substitutions disabled mutation suppression function *in vivo* as shown by elevated Rif$^R$ frequencies for cultures expressing the *recognition-* and *catalysis-* versions. The mutation frequencies for cultures expressing these site-specific substitution variants were comparable to the Rif$^R$ frequencies measured for *null* cultures as judged by overlapping 95% confidence intervals (Fig 8 and S4 Table). These results indicate that the LCHF MutYs suppress mutations by a mechanism that is highly similar to the strategy executed by *Ec* MutY and *Gs* MutY.

## Discussion

To gain insight into DNA repair strategies in early Earth-like environments, we investigated the status of the GO DNA repair system within microbes inhabiting the LCHF. Our approach included mining of metagenomic data, bioinformatic comparisons informed by structure and mechanistic understanding, predictive molecular modeling, and functional analysis. The degree to which this approach succeeded was dependent on the assembly of metagenomic sequences into contigs long enough to contain full-length genes [35]. Earlier attempts to search for MutY genes within previous LCHF metagenomes with shorter contigs yielded a number of hits, but these were truncated and therefore missing critical motifs, explaining weak mutation suppression function (unpublished results). The longer contigs utilized in this study allowed us to capture entire MutY genes, bin these MutY-encoding contigs into MAGs to assess associated gene inventories, and thereby infer metabolic strategies for the microbes expressing the GO DNA repair components.

Within the initial set of 649 LCHF MutY candidates identified by sequence identity, 160 genes encoded proteins with all of the chemical motifs known to be important for MutY function. Indeed, leveraging the extensive body of knowledge obtained from crystal structures and mechanistic studies allowed us to select features such as sequence length, presence of MutY motifs, and structural prediction to distinguish LCHF MutYs from other members of the helix-hairpin-helix (HhH) superfamily. Recombinant expression in *E. coli* revealed that LCHF MutY representatives suppress mutations *in vivo* by a mechanism that depends on the catalytic and OG-recognition motifs (Fig 8), strongly suggesting these are functional enzymes that actively seek and initiate repair of OG:A lesions within their respective LCHF microbes. Toxicity observed for one LCHF gene encoding *Flavobacteriaceae* MutY, which could not be cloned except with disabling nonsense mutations, underscores the risks and dangers posed by MutY and DNA glycosylases in general, which initiate DNA repair by damaging the DNA further, creating AP sites that are themselves destabilizing [60]. The potential for lethal outcomes

makes cross-species function observed for *Marinosulfonomonas* MutY and *Rhodobacteraceae* MutY across vast evolutionary time all the more remarkable.

Retained function across evolutionary and species barriers strongly suggests that MutY interacts with the base excision repair apparatus through some well-preserved mechanism that relies on a universal language understood by all organisms. Most critically, the AP sites generated by MutY should be recognizable to downstream AP nucleases. Protein-protein interactions between AP nucleases and MutY have been discussed as a possible mechanism [61–63], but on its own such a mechanism would rely on coevolution of protein partners. Our results and those reported by others for complementation with the eukaryote homolog MUTYH [64] speak for a mechanism that is less sensitive to sequence divergence. Therefore, we favor a model where the distorted DNA structure created by MutY signals the location of the AP site for handoff to the BER apparatus, as has been suggested previously [62].

*Thiotrichaceae* MutY underperformed in our functional evaluation, despite coming from a gammaproteobacterium most closely related to the *E. coli* employed for the bioassay. Lower mutation suppression performance observed for *Thiotrichaceae* MutY may simply be due to differences in conditions for our *in vivo* experiments and more extreme conditions found in the habitat where *Thiotrichaceae* thrives at the LCHF. In support of this adaptation to extreme environments idea, the LCHF enzymes which were predicted to have the highest stability and form the most persistent ligand complexes in MD simulations appeared incompatible with mesophile biology, being either apparently lethal (*Flavobacteriaceae* MutY) or relatively ineffective at suppressing mutations (*Thiotrichaceae* MutY). This pattern of predicted high stability incompatible with mesophile biology extends also to the reference enzyme *Gs* MutY which is from a known thermophile and also appeared lethal in the reporter bacterium, necessitating a chimera approach for evaluation of biological function [33]. Adapted for stability at higher temperatures, these enzymes may lack flexibility needed to perform their catalytic duty at lower temperatures, an idea described previously as "corresponding states" of conformational flexibility [65–67]. In future work, characterization of LCHF MutY enzymes at high temperatures could address this model directly.

Our metagenomic analysis revealed that gene homologs encoding the GO DNA repair system are abundant in basement microbes inhabiting the LCHF (Fig 3). This observation is surprising given that the basement of the LCHF is expected to be anoxic [1, 3]. The chemical agents commonly thought of for producing oxidized guanine (OG) are ROS derived from molecular oxygen *via* aerobic metabolism. In an anoxic environment, what chemical agents are producing OG and how are these generated? Models of hydrothermal field chemistry predict abiotic production of ROS [68], which the microbial residents may encounter, although these would probably react with cell protective structures before encountering DNA. Continual mixing of seawater with the anoxic hydrothermal fluid could provide molecular oxygen at the interface where hydrothermal fluids vent into ambient seawater at the seafloor [1, 3]. Facultative anaerobes at this interface would inevitably generate ROS [69, 70], and therefore benefit from the GO DNA repair system. Intermicrobial competition has driven acquisition of chemical strategies, including ROS, for killing other bacteria [71–73], but there is currently no evidence for such bacterial warfare in basement dwelling microbes of the LCHF.

Another explanation for the source of OG in basement microbes of the LCHF, which is suggested by our gene inventory analysis (Table 2 and S2 Table), involves reactive sulfur species (RSS) and reactive nitrogen species (RNS). Many of the basement-dwelling microbes within the LCHF appear to metabolize sulfur and nitrogen for energy consumption [4, 35], strategies that generate RSS and RNS as metabolic byproducts [74, 75]. Indeed, mechanisms for the oxidation of guanine by both RSS and RNS have been described, including the formation of 8-oxoguanine (OG) and chemically similar 8-nitroguanine, which templates for adenine in a

fashion similar to OG [76–78]. The oxidation of guanine by RSS and RNS generated from microbial metabolism would produce OG independent of molecular oxygen and thereby necessitate the GO DNA repair system for both facultative and obligate anaerobes inhabiting the LCHF.

Whether organisms developed biochemical systems to deal with oxidative damage before or after the Great Oxidation Event (GOE) remains an open question [79, 80]. It is reasonable to think of these systems arising in response to the selective pressure of oxidative damage from rising $O_2$ levels. However, it is also possible that these systems were already in place as a coping mechanism for other oxidants and were repurposed to deal with the new source of oxidizing agents when $O_2$ became readily available. Indeed, obligate anaerobes contain many of the same pathways to deal with oxidative damage as aerobes [79, 80]. Our discovery of the GO DNA repair system in basement dwellers of the LCHF adds to this body of evidence and supports the hypothesis that oxidative damage repair systems were established before the GOE. We considered the caveat of possible phage-mediated gene transfer–modern microbes adapted to oxygen-rich regions elsewhere may be the source of LCHF MutYs. However, the correspondence of taxonomic assignments based on MutY sequence and based on MAGs and the high degree of sequence diversity seen for LCHF MutY enzymes are inconsistent with expansion of GO DNA repair in the LCHF by horizontal gene transfer. Thus, it seems more likely that LCHF microbes inherited the GO DNA repair system from a common ancestor and retained it through necessity, even in the absence of extrinsic $O_2$.

## Conclusion

Performing empirical studies on how life may have evolved on Earth and other planets is inherently difficult due to time and spatial barriers. Unique sites such as the LCHF serve as representatives of these theorized environments [3, 81]. By discovering the GO DNA repair system at the LCHF and validating mutation suppression function by LCHF MutYs, we infer that microbes within the anoxic environment of the LCHF basement are under evolutionary pressure to repair OG lesions. Evolutionary pressure and the source of OG appear to be driven by nitrogen reactive species or sulfur reactive species as supported by the metabolic survey of the MutY-encoding organisms. These results highlight the need for DNA-based life to manage oxidized guanine damage even in anoxic environments. Moreover, this work adds evidence for the more general hypothesis that life established biochemical systems to deal with oxidative damage early, well before the GOE, and should be considered when developing an evolutionary model for early life.

## Materials and methods

### Metagenomic sequencing and analysis of LCHF fluid samples

Generation, assembly, and annotation of metagenomes from the Lost City Hydrothermal Field (LCHF) have been described previously [35], and are briefly summarized here. In 2018, the remotely operated vehicle (ROV) Jason collected samples of fluids venting from chimneys at the LCHF, which is located near the Mid-Atlantic Ridge at 30 ºN latitude and a depth of ~800 m. Whole-genome community sequences ("metagenomes") were generated from the fluid samples, and assembled metagenomic contigs were binned into metagenome-assembled genomes (MAGs). Potential gene homologs encoding enzymes involved in the GO DNA repair system were identified by conducting KEGG [82, 83] orthology assignment using the *BlastKOALA* v2.2 program [84]. The selected genes that were identified include: *mutT* (KEGG ID: K03574), *mutM* (K10563), and *mutY* (K03575) along with the genes *xthA* (K01142) and *nfo* (K01151), which encode exonuclease III and endonuclease IV, respectively.

The relative abundance of each GO repair pathway gene homolog at each LCHF chimney location was calculated as the normalized metagenomic sequence coverage, determined by mapping of reads from each fluid sample against the pooled assembly. Coverages are reported as transcripts (or metagenomic fragments) per million (TPM), which is a proportional unit suitable for comparisons of relative abundances between samples [35, 85].

## Identification of LCHF MutYs

Candidate MutY genes in the LCHF metagenomes were identified with a *BLASTP* search against predicted protein sequences from the LCHF pooled metagenomic assembly using MutY queries from *Gs* MutY (NCBI Accession ID: P83847.2) and *Ec* MutY (NCBI Accession ID: CDP76921.1). The diversity of these candidates was visualized by aligning the sequences along with *Gs* MutY and *Ec* MutY with *Clustal Omega* [86], and an initial phylogeny was built with *iTOL* [87]. LCHF MutY candidate sequences were aligned by *PROMALS3D* [88], guided by the structure of *Gs* MutY (PDB ID 6u7t). Sequence diversity in the C-terminal domain prevented reliable alignment of this region in this first pass at structure-guided alignment. To overcome this challenge, sequences were split into two parts: one part with all residues before position Val147 in the *Gs* MutY protein which were reliably aligned, and the second part with all residues following position Asn146 which were aligned inconsistently in the first pass. These two parts were separately resubmitted for alignment by *PROMALS3D* guided by the corresponding portions of the crystal structure. For inclusion in this alignment, the C-terminal part was required to pass a minimum length criteria of 160 residues. The resulting alignment was inspected for the MutY-defining chemical motifs described in the text, and a phylogeny was constructed for the 160 authenticated LCHF MutYs with *iTOL* [87]. Selection of the four representative LCHF MutYs was guided by this phylogeny and by the completeness of each associated MAG.

## Taxonomic classification

Contiguous DNA sequences containing the LCHF MutY representatives were assigned taxonomic classifications using the program *MMseqs2* [89] and the Genome Taxonomy Database (GTDB) as described previously [35]. Taxonomic classification of each MAG that included a contig of interest was performed with *GTDB-Tk* v1.5.1 [90]. The environmental distributions of MutY-encoding MAGs were inspected for potential signs of contamination from ambient seawater. This possibility was ruled out by the absence of all MutY-encoding taxa reported in this study in the background seawater samples. MAG completeness and contamination scores were generated by *CheckM* v1.0.5 as described previously [35].

## Prediction of physical parameters

The theoretical molecular weights and pIs of the LCHF MutY representatives and known MutY sequences were generated with ExPASY [91]. The theoretical melting temperature of the representatives was calculated with the Tm predictor from the Institute of Bioinformatics and Structural Biology, National Tsing-Hua University [92].

## Molecular modeling

Protein structures for the LCHF MutY representatives were predicted by *Colabfold* with use of MMseqs2 alignments and relaxed with the *Amber* force field [44, 89, 93, 94]. Predicted structures were superimposed with the crystal structure of *Gs* MutY (PDB ID 3g0q) to generate RMSD (Å) for pruned atom pairs using the MatchMaker tool in *ChimeraX* [45]. Initial

superpositions were dominated by residues in the N-terminal domain. To fairly compare structures for the more diverse C-terminal domains, the linker region between domains was identified by inspection, and superposition with *Gs* MutY was repeated with selection of residues in the N-terminal domain and, separately, in the C-terminal domain.

Ligand docking experiments were executed with the program *AutoDock VINA* [46, 47]. Ligand structures representing adenosine and OG were prepared with the ligand preparation tools implemented with *Autodock Tools* [95, 96]. Receptor structures were prepared from the structures predicted by *Colabfold* or from the crystal structure of *Gs* MutY (PDB ID 6u7t), each after superposition with PDB ID 3g0q, with the receptor preparation tools as implemented with *Autodock Tools* [95, 96]. Receptor structures were treated as rigid objects, and ligands included two active torsion angles defined by the C1'-N9 and C4'-C5' bonds. Separate 24 x 24 x 24 $\text{Å}^3$ search volumes were defined for adenosine and for OG. The adenosine search volume was centered on the position of atom C1' in the residue A5L:18 in chain C of the *Gs* MutY crystal structure (PDB ID 3g0q), and the OG search volume was centered on the position of atom C1' in residue 8OG:6 in chain B of the same structure.

MD simulations were performed with *GROMACS* version 2022.5 [48], applying the *Amber*99SB and GAFF force fields [49, 50], with CPU and GPU nodes at the University of Utah's Center for High Performance Computing. We followed steps outlined in the *GROMACS* tutorial "Protein-Ligand Complex" as a guide for our experiments [97]. The starting structure for a protein-ligand complex was selected from the binding modes predicted by *Autodock VINA*, choosing the mode with the highest affinity after excluding those that appeared incompatible with the double stranded DNA-enzyme structure. To conserve computational resources, simulation of the complex with adenosine was limited to N-terminal residues as follows: residues 8–220 for *Gs* MutY (PDB ID 6u7t); 6–230 for *Marinosulfonomonas* MutY; 2–220 for *Rhodobacteraceae* MutY; 11–223 for *Thiotrichaceae* MutY; and 2–209 for *Flavobacteriaceae* MutY. Simulations with OG omitted the iron-sulfur cluster domain and inter-domain linker and thus introduced chain interruptions as follows: residues 29–137, 234–289, 295–360 for *Gs* MutY; 40–142, 239–352 for *Marinosulfonomonas* MutY; 38–139, 233–352 for *Rhodobacteraceae* MutY; 32–140, 238–364 for *Thiotrichaceae* MutY; and 19–127, 234–354 for *Flavobacteriaceae* MutY. Ligand topology files were generated with the *ACPYPE* server [98], applying the general *Amber* force field [49]. Each complex was solvated with water molecules with three points of transferable intermolecular potential (TIP3P). Counterions were added to neutralize the net charge of the system. The system was energy minimized by 50000 steepest descent steps and further equilibrated in two phases, NVT followed by NPT, each entailing 100 ps with 2-fs steps. Temperature coupling during NVT and NPT equilibration was accomplished with a modified Berendsen thermostat set to the reference temperature 300 K. Pressure coupling during NPT equilibration was accomplished with the Berendsen algorithm set to the reference pressure 1 bar. The equilibrated system was subjected to a 100-ns MD production run with 2-fs steps, applying a modified Berendsen thermostat (300 K reference temperature) and Parrinello-Rahman barostat (1 bar reference pressure). Short range interaction energies, distances, and structures were extracted from the resulting trajectories with use of *GROMACS* functions and plotted with the *R* package *ggplot2* [99]. Figures and movies showing structures were created with *ChimeraX* [45].

## Recombinant DNA cloning

Synthetic genes encoding the LCHF MutYs were codon optimized for expression in *E. coli* except that pause sites with rare codons were engineered so as to retain pause sites found in the gene encoding *Ec* MutY. GBlocks gene fragments were ordered from Integrated DNA

Technologies and cloned into the low-expression pKK223 vector by ligation-independent cloning (LIC). PCR reactions with the high-fidelity Phusion polymerase (Agilent) amplified the synthetic gBlock and two overlapping fragments derived from approximate halves of the pKK223 plasmid. PCR products were separated by electrophoresis in 0.8% agarose x1 TAE gels containing 1 μg/mL ethidium bromide. DNA was visualized by long-wavelength UV shadowing to allow dissection of gel bands, and the DNA was purified with the GeneJet gel extraction system (Thermo Scientific), treated with Dpn1 (New England Biolabs) at 37 ˚C for 45 min, and heat shock transformed directly into DH5α competent cells. Clones were selected on media plates containing 100 μg/mL ampicillin. The plasmid DNA was purified from 4-mL overnight cultures by use of the Wizard Plus MiniPrep kit (Promega) according to the manufacturer's instructions. The sequence of the LCHF MutY encoding gene was verified by Sanger sequencing with UpTac and TacTerm primers. Genes encoding site-directed substitution variants were created by amplifying two overlapping fragments of the LCHF MutY-pKK223 plasmid with mutagenic PCR primers followed by similar gel purification and transformation procedures. In our hands the LIC cloning efficiency was close to 95% except for the *Flavobacteriaceae* MutY encoding gene which could not be cloned intact. The pKK223 plasmids have been deposited with AddGene identifiers 210791–210799 for the LCHF MutY encoding plasmids, 213110 for the *Ec* MutY encoding plasmid, and 213111 for the empty vector.

### Mutation suppression assay

Mutation rates were measured by the method outlined previously [33, 57]. The CC104 *mutm::KAN muty::TET* cells [100] were heat-shock transformed with a pKK223 plasmid encoding the *Ec* MutY gene, LCHF MutY genes, or no gene (*null*). Transformants selected from media plates containing 10 μg/mL kanamycin, 100 μg/mL ampicillin, and 12.5 μg/mL tetracycline (KAT) were diluted prior to inoculation of 2-mL KAT liquid media, and these cultures were grown overnight for 18 hours at 37˚C with shaking at 180 rpm. Cultures were kept cold on ice or at +4 ˚C prior to further processing. Cells were collected by centrifugation, the media was removed by aspiration, and cells were resuspended in an equal volume of 0.85% sodium chloride before seeding 100 μL aliquots to media plates containing 10 μg/mL kanamycin, 100 μg/mL ampicillin, and 100 μg/mL rifampicin (KAR). Dilutions of the washed cells were also seeded to KAR plates ($10^{-1}$ dilution) and KA plates ($10^{-7}$ dilution), and incubated overnight at 37˚C for 18 hours. The number of Rif$^R$ mutants was counted by counting the colony forming units (CFU). Statistical analysis was performed in *R* as previously described [33]. Confidence intervals were obtained by bootstrap resampling of 10,000 trials as implemented in *R* with the *boot* package [101, 102].

### Supporting information

**S1 Fig. MutY gene neighbors.** Structural homology for *Ec* YggX (left) and nearest neighbor to *Thiotrichaceae* MutY (right). The solution NMR structure for *Ec* YggX (PDB ID 1yhd) [103] is superimposable to the structure predicted by *Colabfold* for the nearest neighbor to *Thiotrichaceae* MutY, with RMSD of 0.99 Å for 69 pruned pairs selected from 88 possible pairs. The Cys residue critical for function is highlighted with a yellow sphere for the sulfhydryl group. (PDF)

**S2 Fig. *Colabfold* structure prediction pLDDT scores.** pLDDT scores represent the confidence in the prediction calculated by *Colabfold*. (PDF)

**S3 Fig. Molecular dynamics.** Molecular dynamics simulations were calculated by *GROMACS* with the *Amber*99SB and GAFF force fields for MutY complexed to adenosine and OG. For each MD simulation, short range interaction energies, distances between the ligand and functionally relevant residues, and representative structures sampled after equilibration and at 10,000 ps are shown. Note that the Y-axis is logarithmic for distance. The adenosine and OG ligands are shown with all atoms wrapped in transparent surfaces. For the adenosine complexes, the protein structure was truncated so as to focus on the NTD (residues 8–220 in *Gs* MutY, and corresponding residues for the LCHF MutYs). Catalytic residues are shown: Glu43 and Asp144 in the *Gs* MutY protein and corresponding residues in the LCHF MutYs. The distance versus time plot for the adenosine complex, tracks potential contacts between the catalytic Glu (atoms OE1 and OE2) and the hydrogen bond donors and acceptors on adenosine (atoms N1, N6 and N7). For the OG complex, the iron-sulfur cluster domain and interdomain linker were omitted so as to focus on the OG-recognition site found at the interface between NTD (residues 29–137 in *Gs* MutY) and CTD (residues 234–360 in *Gs* MutY). Residues that interact with OG are shown: Thr49, and Ser308 in the *Gs* MutY protein and corresponding residues in the LCHF MutYs. The distance versus time plot for the OG complex tracks potential contacts between the critical Ser/Thr residues and the hydrogen bond donors and acceptors on OG (atoms N1, N2, O6, N7 and O8). The total short range interaction energy (black trace) is the sum of short range Leanord-Jones (salmon trace) and Coulombic (sky blue) interaction energies. (A) Molecular dynamic simulation for *Gs* MutY NTD complexed with adenosine. The ligand complex persisted for the entire 100,000 ps, with changes in location and orientation evident at 16,000 ps and 42,000 ps in the distance plot. Hydrogen bonds between catalytic Glu43 and the Hoogsten face of the adenine base were observed during the first 16,000 ps. These consistently involved direct contact with N6, as evidenced by close distance (green traces) and inspection of structures. N7 was also engaged (blue traces), with relevance for catalysis, via bridging water molecules (O red and H white). (B) Molecular dynamic simulation for *Gs* MutY complexed with OG. The ligand complex was stable for 92,000 ps, with the OG ligand bound to a cleft between the NTD (white) and CTD (gray). The functionally relevant hydrogen bond between the amide N of Ser308 and atom O8 of OG was frequently observed (not shown), sometimes accompanied by a second OG-specific hydrogen bond between the hydroxyl of Ser308 and atom N7 of OG (sky blue trace in the distance plot). (C) Molecular dynamic simulation for *Marinosulfonomonas* MutY NTD complexed with adenosine. In the first 3,000 ps, the adenine base approached closely catalytic Glu49 (green traces), often directly hydrogen bonded and occasionally bridged by a solvent molecule. However, the complex was relatively unstable, and the ligand departed the active site and found a new binding site by 8,000 ps. Favorable VDW interactions characterize both binding sites, but favorable Coulombic interactions are diminished substantially at the second site. (D) Molecular dynamic simulation for *Marinosulfonomonas* MutY complexed with OG. The initial ligand complex was unstable with a hinge-like motion creating new contacts between the NTD (white) and CTD (gray). After nearly escaping at ~4,000 ps, the OG ligand found several alternate sites on the NTD and CTD. (E) Molecular dynamic simulation for *Rhodobacteraceae* MutY NTD complexed with adenosine. The complex was relatively unstable. The adenine base initially hydrogen bonded with catalytic Glu45 during the first 3,800 ps but then changed orientation and drifted to a new site distinct and different from its original docking site. Note, catalytic Glu45 is not visible in the 10,000-ps representative structure as the new position of adenosine blocks its view. (F) Molecular dynamic simulation for *Rhodobacteraceae* MutY complexed with OG. The ligand complex was unstable and dissociated completely within 48 ns. Functionally relevant hydrogen bonds between Thr299 and OG observed for the initially equilibrated structure were lost as the ligand moved to new positions on the NTD and CTD

before dissociation. (G) Molecular dynamic simulation for *Thiotrichaceae* MutY NTD complexed with adenosine. Note that Ser replaces active site Tyr for this LCHF MutY, as is also the case for *Ec* MutY. The complex was relatively stable with the ligand persisting in the active site throughout the simulation. Hydrogen bonds between catalytic Glu46 and the Hoogsteen face of adenine were evident by close distance to N7 (blue traces) and N6 (green traces) and by inspection of structures. Water frequently bridged N7 to Glu46 as seen in the representative structure at 10,000 ps. (H) Molecular dynamic simulation for *Thiotrichaceae* MutY complexed with OG. The ligand complex was stable for the entire 100,000-ps simulation with the OG ligand bound to a cleft between the NTD (white) and CTD (gray). Hydrogen bonds between Ser306 and OG were frequently observed. (I) Molecular dynamic simulation for *Flavobacteriaceae* MutY NTD complexed with adenosine. The ligand persisted in the active site throughout the simulation, periodically finding new orientations as evident in different distance traces vying for close approach to catalytic Glu33. For example, N7 of the adenine base was very close to Glu33 (blue trace) during the first 2,700 ps, suggesting catalytic engagement, but slipped out of reach at later time points. Water frequently bridged contacts between Glu33 and the adenine base. (J) Molecular dynamic simulation for *Flavobacteriaceae* MutY complexed with OG. The ligand complex was relatively stable. Functionally relevant hydrogen bonds between Ser305 and the Hoogsten face of OG can be inferred from recurring close distances up until 13,000 ps when the ligand adopts a new pose at the NTD-CTD interface.
(PDF)

**S1 Dataset. Alignment of Lost City MutY homologs.** Chemical motifs are highlighted in columns. Alignment was generated by *Promals3D* [88], guided by the structure of *Gs* MutY. It was necessary to align sequences in the first block including up to N146 separately from the second block and third block because otherwise the C-terminal domain residues were aligned inconsistently. The homologs flagged with dark red highlighting were eliminated because of missing chemical motifs. The homolog flagged with light pink highlighting required manual adjustment so as to align the H-x-FSH motif. The representative LCHF MutYs have the following contig ids: *Marinosulfonomonas* MutY, c_000001803648; *Rhodobacteraceae* MutY, c_000002747260; *Thiotrichaceae* MutY, c_000000598175; *Flavobacteriaceae* MutY, c_000001535696.
(PDF)

**S1 Table. Percent identity matrix.** We were interested in determining how similar the amino sequences of LCHF MutY representatives were to existing MutY enzymes. We visualized this in the form of a percent identity matrix that was generated by *Clustal Omega* [86].
(PDF)

**S2 Table. Metabolic gene identification.** [a] *Marinosulfonomonas* MutY contig belongs to two separate MAGs and each are reported separately as MAG 1 and MAG 2, respectively. [b] KEGG ID gene not identified in any MAG and not reported in Table 2. [c] Completeness and contamination scores generated by CheckM v1.0.5 as described in Brazelton et al 2022 [35]. A KEGG ID analysis was used to identify the potential metabolic strategies of the MutY encoding organisms at the LCHF. The full metabolic KEGG ID search is shown above.
(PDF)

**S3 Table. Ligand binding affinity (kcal / mol) [*].** [*] Binding affinities are reported for the binding modes generated by *AutoDock VINA*. Each mode represents a predicted ligand pose, which differs by a combination of position, orientation, and rotamer conformation. The receptor structure was obtained from PDB ID 6u7t for *Gs* MutY and through structure prediction for the LCHF *Marinosulfonomonas* MutY, *Rhodobacteraceae* MutY, *Thiotrichaceae* MutY, and

*Flavobacteriaceae* MutY. The binding mode representing the starting complex for molecular dynamics analysis is highlighted.
(PDF)

**S4 Table. Rifampicin resistance assay.** [a] Confidence intervals (95%) determined by a bootstrap method, see Materials and methods for details. [b] Mutation frequency reported as median number of resistant colonies per $10^8$ viable colonies. Fold change was calculated by dividing Rif$^R$ frequency by the frequency measured for cultures expressing *Ec* MutY.
(PDF)

**S1 Movie. Molecular animations.** Structures for each MD trajectory were sampled at 200-ps intervals from 0–10,000 ps and at 1,000-ps intervals from 10,000–100,000 ps and movies were recorded with *ChimeraX*. Residues belonging to the NTD are depicted with a traditional ribbon style cartoon and colored light gray. Residues belonging to the CTD are depicted with a licorice cartoon style and colored dark gray. Solvent molecules that are within 4 Å of both the ligand and protein are shown (O, red; H, white). Each movie highlights particular features and events with time paused, the scene rotating about the y axis, and a brief caption. Molecular animations may be viewed via the YouTube playlist *MD simulations for LCHF MutYs* at the channel @biochemuu7993.
(PDF)

**S2 Movie. Molecular animation for *Gs* MutY NTD complexed with adenosine.** Adenosine remains within the active site pocket throughout the entire 100,000-ps simulation. At ~45,000 ps adenosine rotates within the active site to place its sugar in close proximity to the catalytic Glu43 residue, demonstrating a limited degree of flexibility within the active site pocket.
(MP4)

**S3 Movie. Molecular animation for *Gs* MutY complexed with OG. OG remains wedged between NTD and CTD for most of the trajectory**. Interactions with the Hoogsteen and Watson Crick face of OG relevant for OG recognition are highlighted at pauses. A new pose emerges at 90,000 ps just prior to departure of the ligand from the NTD-CTD interface.
(MP4)

**S4 Movie. Molecular animation for *Marinosulfonomonas* MutY NTD complexed with adenosine.** Adenosine slips back toward the entrance of the active site pocket within 1ns and exits completely by ~6,000–7,000 ps. It then settles in a pocket on the surface of the protein defined by the loop containing Ser24 and a helix from Ala58 to His65. It remains there until ~25,000 ps when it begins to move freely in the solvent, engaging, disengaging, then re-engaging with the surface of the protein for the remainder of the 100,000-ps simulation.
(MP4)

**S5 Movie. Molecular animation for *Marinosulfonomonas* MutY complexed with OG.** The two domains adopt a different disposition with a new inter-domain interface early in the simulation. The OG ligand finds two new binding sites on the NTD, each distinct from the original binding site, and persists complexed with the NTD until the end of the 100,000-ps simulation.
(MP4)

**S6 Movie. Molecular animation for *Rhodobacteraceae* MutY NTD complexed with adenosine.** Similar to the *Marinosulfonomonas* MutY NTD simulation, adenosine is completely outside the active site pocket relatively early in the simulation by ~5,000 ps. It then settles on the surface of the protein and wedges into a groove with residues Gly126 and Tyr128 on one side and Gln49 and Arg93 on the other side, and remains at this binding site for the rest of the

100,000-ps simulation.
(MP4)

**S7 Movie. Molecular animation for *Rhodobacteraceae* MutY complexed with OG.** The animation features a highly dynamic OG-MutY complex that dissociates completely by 48,000 ps. The OG ligand disengages from functionally relevant interactions at the NTD-CTD interface to find a new site on the NTD by 4,400 ps, nearly escapes at 13,000 ps, and samples several alternate sites on the NTD or the CTD or at a new site at the NTD-CTD interface prior to exiting this region and exploring new sites on the surface of the NTD. The molecular animation is discontinued at 48,000 ps with the complex dissociated. The NTD-CTD structure remains intact for the remainder of the 100,000-ps simulation but the OG ligand did not rebind (not shown).
(MP4)

**S8 Movie. Molecular animation for *Thiotrichaceae* MutY NTD complexed with adenosine.** Adenosine remains in the active site pocket for the entire 100,000-ps simulation. Similarly to the *Gs* MutY-adenosine simulation, the ligand rotates within the active site pocket at ~43,000 ps to place its sugar within close proximity of the active site Glu46 residue, demonstrating limited flexibility within the active site pocket.
(MP4)

**S9 Movie. Molecular animation for *Thiotrichaceae* MutY complexed with OG.** The complex persists for the entire 100,00-ps simulation. The initial complex features interaction of Ser306 with the Watson-Crick-Franklin face of the OG base. This pose persists until transition to a new pose at ~69,000 ps with the deoxyribose sugar closer to Ser306 and the base wedged between two helixes that converge at the NTD-CTD interface.
(MP4)

**S10 Movie. Molecular animation for *Flavobacteriaceae* MutY NTD complexed with adenosine.** Adenosine remains within the active site pocket for the entire 100,000 ps, It starts with its sugar facing the catalytic Glu33 residue. At ~3,000 ps it rotates to bring the base portion deeper within the active site. It remains in this general orientation for the remainder of the 100,000 ps with the sugar engaging Glu33 in the ~60,000–80,000-ps time window.
(MP4)

**S11 Movie. Molecular animation for *Flavobacteriaceae* MutY complexed with OG.** During the first 9,800 ps Ser305 makes hydrogen bonds with the Hoogsteen face of OG in a manner relevant for recognition. At 10,000 ps a new pose emerges with the base wedged between helices in the NTD and CTD and thus removed from the FSH recognition loop. The complex with this new pose persists for the remainder of the 100,000-ps simulation.
(MP4)

## Acknowledgments

Support and resources from the Center for High Performance Computing at the University of Utah are gratefully acknowledged. We thank Markel Kolendrianos, Peyton Russelberg, and Sonia Sehgal for technical expertise. We thank the University of Utah undergraduate students enrolled in Molecular Biology of DNA Lab (BIOL3525 fall 2022) for their contribution to measuring Rif$^R$ mutant frequency data. In particular, we would like to thank the following students for their extra efforts and contributions in the lab: Tieker Duncan, Shilpi Kharidia, Jackson Munn, Kenzie Fleming, Alex Ballinger, Andrew Petersen, Brook Miller, Tom Christensen,

Madison Haught, Emi Wickens, Spencer Sonntag, Abigail Johnston, Sam Aamodt, Jasmine Jacobo, Alyssa Le, Sam Hendry, Saydra Galloway, Hiroshi Aoki, Peyton Merchant, Kaliece Carter, Annie Joseph, Kathleen Brabb, Natalie Morgan, Sophia Khalaji, Helena Haddadin, Hadlee Young, Brenden Roberts, Mason Hansen, Mackenzie Montzingo, Sonia Sehgal and Quyen Tran. We especially acknowledge Emily Dart who first searched LCHF metagenomes for MutY homologs which provided the impetus for development of this project.

## Author Contributions

**Conceptualization:** Payton H. Utzman, Vincent P. Mays, William J. Brazelton, Martin P. Horvath.

**Data curation:** Payton H. Utzman, Vincent P. Mays, Briggs C. Miller, William J. Brazelton, Martin P. Horvath.

**Formal analysis:** Payton H. Utzman, Vincent P. Mays, Briggs C. Miller, William J. Brazelton, Martin P. Horvath.

**Funding acquisition:** William J. Brazelton, Martin P. Horvath.

**Investigation:** Payton H. Utzman, Vincent P. Mays, Briggs C. Miller, Mary C. Fairbanks, William J. Brazelton, Martin P. Horvath.

**Methodology:** Payton H. Utzman, Briggs C. Miller, William J. Brazelton, Martin P. Horvath.

**Project administration:** William J. Brazelton, Martin P. Horvath.

**Resources:** William J. Brazelton, Martin P. Horvath.

**Software:** Vincent P. Mays, Briggs C. Miller, William J. Brazelton, Martin P. Horvath.

**Supervision:** William J. Brazelton, Martin P. Horvath.

**Validation:** Payton H. Utzman, William J. Brazelton, Martin P. Horvath.

**Visualization:** Payton H. Utzman, Vincent P. Mays, Briggs C. Miller, William J. Brazelton, Martin P. Horvath.

**Writing – original draft:** Payton H. Utzman, Vincent P. Mays, Briggs C. Miller, William J. Brazelton, Martin P. Horvath.

**Writing – review & editing:** Payton H. Utzman, Vincent P. Mays, Briggs C. Miller, Mary C. Fairbanks, William J. Brazelton, Martin P. Horvath.

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
