## [Decision Letter · Decision Letter 0]

6 Jul 2023

PONE-D-23-10070Metagenome mining and functional analysis reveal oxidized guanine DNA repair at the Lost City Hydrothermal FieldPLOS ONE

Dear Dr. Horvath,

Thank you for submitting your manuscript to PLOS ONE. After careful consideration, we feel that it has merit but does not fully meet PLOS ONE’s publication criteria as it currently stands. Therefore, we invite you to submit a revised version of the manuscript that addresses the points raised during the review process.

We look forward to receiving your revised manuscript.

Kind regards,

Pramodkumar Pyarelal Gupta, PhD

Academic Editor

PLOS ONE

Journal Requirements:

"This work was supported by National Science Foundation (https://www.nsf.gov/) awards to Martin. P Horvath (CHE:CLP- 1905249, 2204229) and to William J. Brazelton. (OCE-1536405). This work was also supported by the NASA Astrobiology Institute Rock-Powered Life team."

3. Please amend your manuscript to include your abstract after the title page.

Additional Editor Comments:

The author should confirm the similarity and identty between the query sequence and database sequences in BLAST research

In case of protein structure prediction, how the authors have assessed the modeled strucrture quality, it should be framed appropriatey and added in the text.

As the Authors have performed molecular docking experiment, what was the rational to select the binding site region, kindly add it.

The molecular docking is not the wholesome validation, the authors should carry out MD Simulation study, which will give us more detail outcome in terms of dynamicity and stability.

Spell check and grammar needs to corrected

The PDB-id should be well define.

Reviewers' comments:

Reviewer's Responses to Questions

**Comments to the Author**

1. Is the manuscript technically sound, and do the data support the conclusions?

Reviewer #1: Yes

2. Has the statistical analysis been performed appropriately and rigorously? 

Reviewer #1: Yes

3. Have the authors made all data underlying the findings in their manuscript fully available?

Reviewer #1: Yes

4. Is the manuscript presented in an intelligible fashion and written in standard English?

Reviewer #1: Yes

5. Review Comments to the Author

Reviewer #1: This well-written and enjoyable manuscript by Utzman et al provides compelling evidence that organisms evolved ways to deal with oxidative DNA damage prior to the Great Oxidation Event. The presence of homologues of the GO DNA repair system in microbes recovered from the Lost City Hydrothermal Field (LCHF) is interesting, and the ability of distant mutY homologues from this environment to complement E. coli mutY mutants is remarkable. This work raises a number of interesting possibilities concerning the importance of ROS and oxidative damage during the evolution of life, even in the absence of high oxygen levels.

I have only a couple of minor notes:

1. In the Discussion, when covering potential sources of ROS/RNS/RSS, beyond being by-products of some metabolic pathways, might these also be generated as a form of inter-bacterial competition or warfare? The idea that many antibiotics act by generating ROS directly or indirectly is gaining traction and might be relevant selective pressures in mixed microbial communities. Also of potential relevance is the idea that some quorum sensing molecules might be progenitors of certain antibiotics, having antibiotic activities against competitors. Is there any evidence of inter-microbial competition/warfare within the LCHF or similar environments?

2. Line 174: '…and the catalytic Tyr residue is often replaced...' (missing ‘d’)

6. PLOS authors have the option to publish the peer review history of their article (what does this mean?). If published, this will include your full peer review and any attached files.

Reviewer #1: No

---

## [Author Response · Author response to Decision Letter 0]

1 Dec 2023

Responses from the authors are provided for each point raised in the initial review. 

The authors have carefully reviewed the PLOS ONE style requirements, including those for file naming, and the revised version matches these style requirements as far as we know.

The funding statement as been amended as requested, added to the cover letter, and this statement is also provided here. Thank you for updating the online submission form. That is a big help. 

Updated Funding Statement:

This work was supported by NSF awards to MPH (CHE:CLP- 1905249, 2204229) and to WJB (OCE-1536405), by the NASA Astrobiology Institute Rock-Powered Life team, and by UROP funding from the Office of Undergraduate Research at the University of Utah awarded to PHU. The support and resources from the Center for High Performance Computing at the University of Utah are gratefully acknowledged. There was no additional external funding received for this study. 

End of Updated Funding Statement

The Abstract is now included after the title page.

Supplementary Information Table S2 reports the pairwise identity among query and database sequences. The text clarifies the % identity cutoff for finding MutY candidate genes in the metagenome and the E-value threshold (see Lines 162-163). Additionally, observations regarding % identity between candidate genes and query sequences are included in the text (see Lines 234-239)

Table 3 now reports the pLDDT scores for each predicted structure. Notes for assessing modeled structure quality have also been added to the text (see Lines 333-348)

The text now describes the rationale for selection of the binding site region in the Results section (see Lines 356-361)

Thank you for this suggestion to validate the molecular docking outcomes. The report now includes MD simulation to analyze the docking outcomes. Interaction energies averaged over a 10-ns simulation are reported in Table 3 for each ligand-receptor complex. The text also describes these outcomes in a new subsection Molecular dynamics simulations (see Lines 366-437). Supporting Information Fig S7 shows time-evolution structures along with time-dependent distance and energy plots. 

Spelling and grammar have been corrected. 

PDB-ids have been defined. 

Thank you for reviewing our manuscript favorably!

Thank you for this interesting observation and suggestion. We have added note in the Discussion to highlight the possibility of bacterial warfare (see Lines 561-563); however, so far no evidence for inter-bacterial competition has emerged from analysis of these communities.

Thank you for pointing out this typographical error which has been corrected (see Line 191). Other mistakes in spelling and grammar have also been corrected.

---

## [Editor Report · Decision Letter 1]

27 Dec 2023

PONE-D-23-10070R1Metagenome mining and functional analysis reveal oxidized guanine DNA repair at the Lost City Hydrothermal FieldPLOS ONE

Dear Dr. Horvath,

Thank you for submitting your manuscript to PLOS ONE. After careful consideration, we feel that it has merit but does not fully meet PLOS ONE’s publication criteria as it currently stands. Therefore, we invite you to submit a revised version of the manuscript that addresses the points raised during the review process.

We look forward to receiving your revised manuscript.

Kind regards,

Pramodkumar Pyarelal Gupta, PhD

Academic Editor

PLOS ONE

**Additional Editor Comments:**

MD simulation which is an important part to the research work, but here the time consideration only for 10-nano second i.e. a very small time limit. Again the authors have mentioned in result and supportive data that the study outcome is unstable in majority of cases.

As a suggestion the time limit should be minimum 100 ns or more, which will give us more data to analyze it in depth.

---

## [Author Response · Author response to Decision Letter 1]

1 Apr 2024

The MD simulations have been extended to 100 ns. The trajectories are summarized as Energy and Distance plots in Supplementary Information Fig S7 and also as molecular movies Supplementary Information Movies. The main text and text for the supplementary information have been updated to include descriptions for the more extensive MD analysis.

---

## [Editor Report · Decision Letter 2]

17 Apr 2024

Metagenome mining and functional analysis reveal oxidized guanine DNA repair at the Lost City Hydrothermal Field

PONE-D-23-10070R2

Dear Dr. Martin P. Horvath

We’re pleased to inform you that your manuscript has been judged scientifically suitable for publication and will be formally accepted for publication once it meets all outstanding technical requirements.

Kind regards,

Pramodkumar Pyarelal Gupta, PhD

Academic Editor

PLOS ONE

Additional Editor Comments (optional):

The MD simulation study is good enough.
---

## [Editor Report · Acceptance letter]

26 Apr 2024

PONE-D-23-10070R2 

PLOS ONE

Dear Dr. Horvath, 

I'm pleased to inform you that your manuscript has been deemed suitable for publication in PLOS ONE. Congratulations! Your manuscript is now being handed over to our production team.

Kind regards, 

on behalf of

Dr. Pramodkumar Pyarelal Gupta 

Academic Editor

PLOS ONE